

# Standardization of a geo-referenced fishing dataset for the Indian Ocean Bigeye Tuna, *Thunnus obesus* (1952-2014)

Teja A. Wibawa[1,2], Patrick Lehodey[1] and Inna Senina[1]

[1] Marine Ecosystem Department, Space Oceanography Division, CLS, 8-10 rue Hermès, 31520 France
[2] Institute for Marine Research and Observations IMRO, Prancak Jembrana Bali 82218 Indonesia

*Correspondence to*: Teja A. Wibawa (twibawa@cls.fr)

**Abstract.** Geo-referenced catch and fishing effort data of the bigeye tuna fisheries in the Indian Ocean over 1952–2014 were analysed and standardized to facilitate population dynamics modelling studies. During this sixty-two years historical period of exploitation, many changes occurred both in the fishing techniques and the monitoring of activity. This study includes a series of processing steps used for standardization of spatial resolution, conversion and standardization of catch and effort units, raising of geo-referenced catch into nominal catch level, screening and correction of outliers, and detection of major catchability changes over long time series of fishing data, i.e., the Japanese longline fleet operating in the tropical Indian Ocean. A total of thirty fisheries were finally determined from longline, purse seine and other-gears data sets, from which 10 longline and four purse seine fisheries represented 96% of the whole historical catch. The geo-referenced records of catch, fishing effort and associated length frequency samples of all fisheries are available at https://doi.pangaea.de/10.1594/PANGAEA.864154.

## 1 Introduction

Bigeye tuna is one of the most valuable tropical tuna species exploited in the Indian Ocean by industrial longline fleets since the 1950s and by purse seine fishery since 1980 (IOTC, 2015; Miyake et al., 2004; Sharma et al., 2014). During 1952-2014, over four million tonnes of bigeye tuna were removed from the Indian Ocean, 74% of it by longline fishing. The longline fishery historically developed by expansion of the Japanese fishery from 1952, after releasing virtual lines set in place at the end of the 2nd World War restricting its fishing activity to Japanese waters only (Haward and Bergin, 2001; Miyake et al., 2004; Okamoto et al., 2004). In the early period, Japanese longline concentrated in the Eastern of Indian Ocean (Menard et al., 2007; Mohri and Nishida, 1999). A few years later, bigeye tuna became also exploited by longline fleets from Korea in 1965 and Taiwan-ROC in 1967 (Miyake et al., 2004). Since then, thirteen longline fleets have been declared fishing bigeye tuna within Indian Ocean. They are: Seychelles, China, Australia, France-La Reunion, South Africa, Mauritius, Thailand, Portugal, France-Mayotte, Maldives, Malaysia, India and Philippines.

More recently, purse seine fishing has become responsible of a significant percentage of bigeye tuna catch in the Indian Ocean, especially in the juvenile age classes, in contrast with longline fisheries targeting adult fish (IOTC, 2015). These surface fisheries started operations in 1980s, when the French purse seine fleet moved from the eastern Atlantic Ocean to the Indian Ocean (Allen, 2010; Majowski, 2007). Target species of purse seiners are skipjack (SKJ) and yellowfin (YFT) for the canning industry, but bigeye tuna (BET) were also caught in small



proportions in the early period of exploitation, when purse seine vessels operated mainly in association with tuna schools (free swimming schools: FS). With the introduction of the fish aggregating device (FAD) fishing technique in the 1990s, the purse seine catch of juvenile bigeye tuna increased significantly, representing nearly half of total bigeye catch in the recent years (Davies et al., 2014; Fonteneau et al., 2013; IOTC, 2015; Kaplan et

al., 2014).

Intensive exploitation by longliners and increasing fishing mortality of juveniles in the last two decades by purse seiners fishing on FADs have reduced bigeye tuna stock in the Indian Ocean to a level close to its maximum sustainable yield (IOTC, 2015). However, the uncertainty on the stock assessment studies for this species is substantial and needs to be reduced by improving both datasets and models. The secretariat of the Indian Ocean

Tuna Commission (IOTC) is collecting and publishing fishing data (catch, effort and size frequency of catch) allowing stock assessment analyses and estimations of fishing mortality. There are two data sets providing nominal and geo-referenced data. The first one gives total annual catch of bigeye tuna by country and gear, while the second provides monthly geo-referenced catch and effort. However, there is some discrepancy between nominal and geo-referenced catch data, the latter being a sub-sampling of the aggregated total catch by Country.

Most stock assessment studies have used nominal catch. New modelling approaches however, require to spatially disaggregating the fishing data either between a few large geographical regions (e.g., Multifan-CL, Hampton and Fournier, 2001), or at a spatial resolution of 1 to few degrees (e.g., SEAPODYM, Lehodey et al., 2015; APECOSM, Dueri et al., 2012). These higher resolution data sets are also used for investigating species habitats, allowing CPUE standardization and more generally the relationships between the species distribution

and the variability of climate and environment. These studies require standardized data sets of historical catch data allowing results inter-comparisons. In the present study, we revise the historical geo-referenced data of bigeye fishing in the Indian Ocean using a careful screening, standardization and validation approach of the main fleets to provide a spatially explicit estimate of catch, effort and length frequency data available from the publically available IOTC database (www.iotc.org).

There are many problems with such long time series of data due to changes in fishing practices and data reporting. The data sets are constructed from various spatial resolutions ranging between 1°x1° and 10° x 20°. Catch and effort data are derived from various types of fishing gears characterized by different fishing methods and target species. Consequently, the catchability, a key coefficient that links the catch to fishing effort and fish abundance, varies from one fishing mode to another. Over time, a variety of catch and effort units were used that

prevent long time series analyses. Finally, for studies requiring computing total fishing mortality, the geo-referenced catch data needs to be raised to match the nominal catch.

Therefore, the objective is to provide a standardized dataset, with a definition of the longline, purse seine and other-gears fisheries, to researchers from various disciplines that may have not the necessary expertise in fisheries sciences to interpret these data correctly. Several steps are described and include the homogenization of

spatial resolutions, the standardization of catch unit, the raising of catch data to fit the total nominal catch, the standardization of effort unit and finally, the analysis of data time series and fisheries history to detect major changes in catchability.

Standardized datasets resulting from this study are provided under ASCII format on PANGAEA (https://doi.pangaea.de/10.1594/PANGAEA.864154).



## 2 Material and method

### 2.1 Data

Nominal and geo-referenced data (catch, effort and size frequency of catch) are freely available on the IOTC website (http://www.iotc.org/data/datasets). The catch and effort data were classified accordingly to three groups
of gear type: longline, purse seine, and other-gears. Bigeye tuna data were extracted from each group and analyzed separately. A data screening of each group using a topographic mask led to excluding 6.87% of longline, 1% of purse seine and 0.97% of other-gears data with wrong position located on land (i.e. all the cell at a given resolution is on land).

#### 2.1.1 Longline

Initial geo-referenced dataset of longline bigeye tuna catch included five categories: longline targeting bigeye tuna (LL), longline targeting swordfish (ELL), fresh tuna longline (FLL), exploratory fishing longline (LLEX) and longline targeting shark (SLL). The last one was ignored since it contained only five observations. In the four remaining, 93% of bigeye tuna catch was due to LL category including Japanese, Taiwanese, Korean, Seychelles, Chinese, Thailand, Mauritius, Maldives, and Philippine fleets. The ELL contributed to 4.4% of
longline bigeye catch data by Australian, French-La Reunion, Seychelles, South African, Portuguese, French-Mayotte and Mauritius fleets. The FLL fleets of Taiwan, China and Malaysia caught the remaining 2.6%, and the LLEX contained only a few Indian longline data (0.05% of total longline data).

The vast majority of longline data (92.68%) were structured in 5x5 degree grid cells. The remaining data were at resolution 1°x1° (7.30%) and 20°x10°. The Maldives-LL and Mauritius-ELL provided all their data at resolution
1°x1°. Several fleets (La Reunion-ELL, Indian-LLEX, Mauritius-LL, Seychelles-ELL, Thailand-LL and South African-ELL) provided 5°x5° data in certain years and 1°x1° in other ones. The 20°x10° cell consisted only of the Mayotte fleet.

IOTC provides two types of bigeye tuna catch unit: total-weight and numbers of individuals. Four categories can be differentiated: catch declaration only in numbers (Japanese-LL: 41.7%), both in total weight and numbers for
the same period (Taiwanese-LL and FLL: 34.8%), total weight and numbers for different periods (Korean-LL and La Reunion-ELL: 13.2%), or alternatively only in weight (all remaining fleets: 10.3%).

Effort units are expressed in number of hooks (99.3% of data), number of fishing days (0.58%) and number of sets (0.16%). The French-Mayotte-ELL and the Philippine-LL used only fishing days. Thailand–LL reported in fishing days in 2013-2014 but also in number of sets in 2007-2008 and in number of hooks in 2011. Portuguese-
ELL declared effort unit in number of hooks in 2008, 2010 and 2011 but also in number of fishing days in 2006-2007 and 2013-2014.

#### 2.1.2 Purse seine

Geo-referenced purse seine fishing was divided between large (PS), and small (PSS) purse seiners with geo-referenced data from fleets of Spain, France, Seychelles, Japan, Mauritius, Thailand, Korea, former Soviet
Union, NEIPS and NEISU. The NEIPS data was collected by European scientist onboard non-European vessels, while the NEISU was collected by Russian scientists from purse seine vessels of Liberia, Belize and Panama. The small purse seine data consisted only of Indonesian observations. Almost all (> 99.9%) purse seine fishing



data were at the resolution of 1°x1°, and a very few data had a resolution of 5°x5°. These data were sub-divided between sets on free schools (FS), associated to artificial (FAD) or natural logs (LS), mixed strategy (MIX) and unknown sets (UNCL). There were 11% of sets purely on free school and 70.8% associated to logs. A very few data (<0.2%) for small purse seiners were reported as unknown. The remaining (17.9%) consisted of large purse

seiners operating either on free school or log but reporting a single fishing effort without distinction of the fishing strategy. The purse seine fishery is dominated by Spanish and French fleets that together provide 65.9%, 65.5% and 59.5% of respectively FS, LS and MIX sets.

Catch of the purse seine data were uniformly expressed in total-weight. The number of fishing hours (FHOURS) is the most used unit of fishing effort (87.3%), followed by the number of fishing days (FDAYS, 7.6%), and the

number of days at sea (DAYS, 4.0%). A very few records (1.1%) used number of sets (SETS) or number of trips (TRIPS). The fishing effort unit can change for a same fleet over certain periods of time. The Spanish fleet reported effort in FDAYS until 1990 but in FHOURS after this year. Similarly, three periods occurred for the Japanese fleet with effort in days at sea (1989-1999), fishing days (2000-2010) and sets (2011-2014). The Thailand fleet had only two years of data with 2006 in fishing days and 2009 in sets.

### 2.1.3 Other-gears

The other fishing gears to which are associated bigeye tuna catch are coastal longline (LLCO) in Maldives, gillnet (GILL) from Taiwan fleet, a combination of gillnet and longline (GL) used in Sri Lanka, hand line (HAND) and baitboat (BB) both used in Maldives and Australia, troll line (TROL) from Maldives, Australia and Indonesia, hand line and troll line (HATR) from La Reunion and Australia, and sport fishing (SPOR) in South

Africa. Some records from Sri Lanka have unknown gear (UNCL). From these various categories, the coastal longline and gillnet represented respectively 53.5% and 23.6% of all records. Spatial resolutions used were either 1x1 (62.2%) or 5x5 (37.8%) degree grid cells, with the lower resolution used by Taiwan-GILL, La Reunion-HATR, Sri Lanka-GL, Sri Lanka-UNCL, South Africa-SPOR and Indonesia-TROL.

Catch of this group were declared in total weight and the fishing effort composed of various units: number of

hooks (HOOKS), number of days with the net in the water (NETS), number of fishing days (FDAYS), number of trips (TRIPS), number of boats (BOATS) and number of days at sea (DAYS).

### 2.1.4 Length frequency

IOTC maintains also a database of length frequency of catch collected onboard fishing vessels by observers or during landing operations. These data provide key information for population dynamics model. The length-

frequency catch data are aggregated either monthly or quarterly. We standardized them all on a quarterly basis. All bigeye tuna size samples were measured in cm fork length (FL). The original size data were distributed in 150 classes starting at 10-cm length with 2-cm intervals between each class. In this study the maximum size was limited to 200cm, a limit used in most stock assessment studies (IOTC, 2015; Langley et al., 2013), since there are only a very few fish caught with bigger size. These size frequencies of catch were associated to their

corresponding fisheries.

Spatial resolution included both regular and irregular cells. Regular cells can be 1°x1°, 5°x 5°, 10°x 10°, or 10°x 20°, with one latitude and longitude position providing the reference corner defined by IOTC as the closest corner to the intersection of zero degree of latitude and zero degree of longitude. Irregular cells cover the



Western Indian Ocean (code area: F51; west of 80° E), the Eastern Indian Ocean (F57; east of 80° E), the Indian Ocean Northwest (IONW; west of 80° E and north of equator), the Indian Ocean Northeast (IONE; east of 80° E and north of equator), the Indian Ocean Southwest (IOSW; west of 80° E and south of equator), and the Indian Ocean Southeast (IOSE; east of 80° E and south of equator).

5  **2.2 Standardization of spatial resolution**

The main spatial-resolution used for geo-referenced catch and effort declaration is 5°x 5° for longline and 1°x1° for both purse seine and other-gears data. These resolutions were selected as representative of these three types of fishery, and data that were not provided at these resolutions were converted to these respective reference spatial resolutions, either by aggregating catch and effort when resolution was higher, or conversely dividing the catch and effort equally in the case of original lower resolution. All longitude and latitude references were adjusted to the centre of each cell.

**2.3 Conversion of longline catch unit**

Length-frequency data were used to convert into total-weight a few Japanese-LL and Korean-LL fishing data detected as outliers (cf below) and thus aggregated in a separate "outlier" data file. Similar conversion was used for the catch of La Reunion-ELL fleet that was reported in numbers. The length frequency data were standardized into monthly 5x5 degree grid cells. Then, it was transformed into weight by using the length-weight relationship $w = aL^b$, with w = weight (kg), a = 1.592 x $10^{-5}$, L = fork length (cm), and b = 3.042 (IOTC, 2013). When spatio-temporal occurrences of catch and length data did not exist, the resolution was decreased over space and time until a length frequency sample can be found and used for the conversion to weight.

20  **2.4 Raising spatial catch and effort to nominal catch level**

The IOTC geo-referenced data set provides a subset of the total catch declared as nominal catch by each country. To compute total fishing mortality from geo-referenced fishing data, this catch needs to be raised to the level of nominal catch. This is a data processing step sometimes conducted directly by National fisheries statistics services before to be provided to the RFMOs (Fonteneau et al., 2013). When the difference between total annual nominal and geo-referenced catch was above 5%, we used a raising factor $I$ and added the product of $I$ with the annual catch difference to the monthly catch of geo-referenced cell $i, j$. The factor $I$ used to distribute the total annual catch differences was computed for each fleet and gear type using the following equation:

$$I_{c,m,ij} = \frac{C_{ij,m}}{\sum C_{ij,m}}$$  (Eq. 1)

where $C_{i,j,m}$ is the catch in the cell of indices i,j of a given month $m$.

30

The same approach and factor was used to raise the fishing effort associated to the catch $C_{i,j,m}$.

Unlike in geo-referenced data, the nominal catch data for purse seiners did not discriminate between type of sets (i.e., FS or LS). To maintain this key information in the geo-referenced dataset the difference between total



annual nominal and geo-referenced catch data were divided proportionally to the proportion of each set type available in the geo-referenced dataset.

The raising procedure did not concern the Japanese-and Korean longline catch data expressed in number of fish (see below).

**2.5 Detection and correction of outliers**

An outlier screening based on the Hampel Identifier method (Pearson, 1952) and using Catch Per Unit of Effort (CPUE) was conducted for each sub-dataset characterized by the same gear, flag, and catch and effort units. Outliers were defined on the basis of a threshold value $t$. A CPUE $x_k$ is defined as outlier if:

$$|x_k - x| / S \geq t \tag{Eq. 2}$$

where $x = \text{median}\{x_k\}$

and $S$ is the scale estimate from the Median absolute deviation from median (MADM)

$$S = 1.486 \, \text{median} \{|x_k - x|\} \tag{Eq. 3}$$

The threshold value was adjusted for each sub-fleet to avoid excessive removing; practically no more than ~5% of each sub-fleet dataset. Following the robust procedure proposed by Davies and Gather (1993), this method was used within a loop until no outliers remained in the dataset. For CPUE records detected as outliers, the effort was corrected relatively to the mean local CPUE of the neighbouring non-outlier observations, with the condition that they occurred at the similar month within a defined maximum radius. An iterative algorithm allowed to selecting the first two adjacent non-outlier CPUE values to compute the local mean CPUE. When the neighbouring observations were not available within the defined radius, the outlier record was moved to a separate fishery where only catch values are retained. This approach was chosen to avoid a loss of information on the total catch. It is preferable to modify the effort, because its value does not influence directly the stock variation (Maunder and Punt, 2004; Maunder et al., 2006).

**2.6 Standardization of effort units**

When possible, fishing effort units were converted to the reference units, i.e., number of hooks for longline and number of fishing hours for purse seine. This was possible when the different units used by a fleet included also the reference unit. In that case, the conversion was based on ratio calculated from mean CPUE of reference and targeted period. When there was no reference unit, the reference was obtained from another fleet with similar characteristics (i.e., similar fishing gear and tuna target). As for the conversion of catch units, when spatio-temporal occurrences of effort in both units did not exist at the original resolution, it was sought by testing decreasing resolution and eventually by using a monthly climatological value.

**2.7 Time series analysis to detect major changes in Japanese longline fishery**

Over the historical industrial fishing period since the 1950s, changes in tuna fishing technologies and tuna market demand have significantly modified the fishing strategy of longline fleets. Introduction of monofilament for the mainline allowing deeper longline sets (Okamoto and Shono, 2006; Okamoto et al., 2001), installation of



super-cold freezers for fish storage (Haward and Bergin, 2001; Matsumoto et al., 2013; Okamoto and Shono, 2006; Ward and Hindmarsh, 2007), and increasing demand for sashimi market (Miyake et al., 2004; Sakagawa et al., 1987) have led to stronger targeting of bigeye tuna.

These changes affected particularly the Japanese longline fleet that has the longest periods of exploitation and
the largest market demand (Haward and Bergin, 2000; Lee et al., 2005; Yeh and Chang, 2013). Consequently, the catchability of the fishing gears and thus the CPUE were modified over time (Fonteneau et al., 2000; Maunder et al., 2006). Therefore, spatio-temporal variability of the Japanese longline CPUE time series was analyzed using a spatial stratification into eight large regions as proposed for stock assessment of Indian Ocean bigeye tuna (Kolody et al., 2010) and CPUE's standardization (Matsumoto et al., 2015) studies, but with
extended north and south boundaries to include all longline data.

Abrupt changes in temporal trends of CPUE were sought using the Breaks for Additive Seasonal and Trend (BFAST) method, which is widely applied for detection of long-term changes (Forkel et al., 2013; de Jong et al., 2012; Lambert et al., 2013; Verbesselt et al., 2010a, 2010b, 2015; Watts and Laffan, 2014). BFAST differentiates a time series ($Y_t$) into a sum of its seasonal ($S_t$), trends ($T_t$) and residual ($e_t$) components. A break is
defined when the slopes in the trends of adjacent periods are significantly different (de Jong et al. 2012). The BFAST method requires defining one parameter; either the minimum duration of the time series before a potential break or the maximum number of breakpoints allowed to be detected within the time series (de Jong et al., 2012). Both approaches were tested in this study. Since BFAST cannot accommodate missing values within time series data, they were replaced by monthly climatological (monthly average) CPUEs when only a few of
them were missing, otherwise the time series was cut.

### 3 Results

The majority of bigeye tuna landings in the Indian Ocean is provided by industrial longline (74.2%), followed by purse seine (18.5%), and other-gears (7.3%). The Taiwanese, Japanese and Korean longline fleets together captured 68% of longline catch. The nominal catch data indicates that the Japanese started to capture bigeye tuna
in 1952, followed by the Taiwanese in 1954, and the Korean in 1965. The catch were largely due to the Japanese fleet during 1952 to the mid-1970s. Then Korea until the mid-1980s, and Taiwan-ROC became two other major actors of the longline fishery (Fig. 1). Finally, the Indonesian fresh longline and the NEI.FROZEN longline fleets contributed respectively to 11% and 5.7% of longline catch, but their geo-referenced catch data are unavailable. The nominal catch of the latter fleet were estimated by IOTC secretariat from various non-reporting
longline flags, including Honduras, Belize, Equatorial Guinea and Panama (Fig. 1).

Catch by industrial purse seiners were dominated by two European fleets: the Spanish (33.7% of purse seine catch) and the French (25.2%). From the early 1980s to the mid-1980s, the French fleet dominated this fishery, then until the mid-1990s the catch from both fleets were at similar level. Since then, the Spanish contributed to the largest annual bigeye tuna catch for this fishing gear (Fig. 1). The Indonesia small purse seine, the Seychelles
and the NEIPS (Sect. 2.12) contributed respectively to 13.8%, 9%, and 7.2% of purse seine catch (Fig. 1). Unfortunately, nominal catch of the Indonesian fleet were not accompanied by geo-referenced data sets.

In the other gears group, more than 80 percent of bigeye tuna landings were provided by six fleets: Indonesia coastal longline (38.2%), Sri Lanka coastal longline (14.9%), Indonesia gillnet (9%), Indonesia liftnets (8%),




Maldives baitboat (6.3%), and Indonesia troll lines (5.8%)(Fig. 1). The geo-referenced data sets of these fleets are unavailable, except for the Maldives baitboat.

### 3.1 Standardization of spatial resolutions

A limited amount of catch and effort data of the longline fleets with resolution at 1°x1° were converted to the

5°x5° resolution common to all longline fishing data. This concerned La Reunion-ELL (2009-2014), Indian-LLEX (2012-2013), Maldives-LL, Mauritius-ELL, Mauritius-LL (2001), Seychelles-ELL (1995-2014), Thailand-LL (2008, 2012, 2013), South African-ELL (2007-2009 and 2011-2014), and Mayotte ELL. Conversely, catch and effort data of several small purse seine and other gear fleets were redistributed over the 1°x1° grid that is common to all fleets defined by these gears. This concerned Indonesian-PSS, Taiwanese-GILL,

La Reunion-HATR, Sri Lanka-GL, Sri Lanka-UNCL, Indonesian-TROL and South African-SPOR (1991, 1992, 1995).

Over the whole period of exploitation, the longline bigeye catch distribution has covered all the Indian Ocean basin up to 50° S but with the maximum catch coming from the tropical region 10° N-15° S (Fig. 2). For the purse seine fishery the catch was also concentrated in the tropical region, but more particularly in the western

Indian Ocean (Fig. 3). The other-gears group concentrated their activities in the central and southern of Indian Ocean. Coastal longline and unknown gears captured bigeye tuna in the central Indian Ocean, and they together contributed to 64.3% of other-gears catch. While bigeye tuna catch from gillnet (32.6% of other-gears catch) are distributed in the southern Indian Ocean (Fig. 3).

### 3.2 Longline catch unit

All Japanese-LL and 75% of the Korean-LL catch data are expressed in number of individual tuna. Rather than introducing potential errors trying to convert number of fish into weight, these data were kept unchanged. A simple test of conversion of individual fish to weight gave a total catch above the nominal catch.

The individual to weight conversion was used for the portion (30%) of catch data of La Reunion-ELL fleet that were declared in number of individual fish.

### 3.3 Data raising to nominal catch

Geo-referenced fishing data from twelve longline fleets, five purse seine fleets and five other gear fleets required to be raised to the nominal catch level (Table 1). The differences vary from year to year as illustrated for the Korean longline fleet (catch in tonnes) in Fig. 4. There are still some periods where in absence of geo-referenced catch, the raising procedure cannot be used leading to underestimated catch, e.g., in 1998 and 2009 of the Korean

LL fleet (Fig. 4).



### 3.4 Outliers

#### 3.4.1 Longline

A total of 2,571 outliers were detected from the longline data set. Using a threshold classically fixed to a value of three, the Japanese-LL fleet (31,963 records) and the Taiwanese-LL fleet (24,918 records) contributed to 2/3 of

this total with respectively 1,218 and 491 outliers (Table S1). Fishing effort of 2,359 outliers (i.e., 91.7% of the total) were corrected using a maximum radius of 15 degrees from the position of the outliers. For 53.4% and 47.8% of the corrected outliers in the fleets using respectively number of individuals or total weight as catch units, a radius of 5° was sufficient to estimate the mean CPUE from neighbouring points. But for a few outliers, it was not possible to correct the fishing effort and then the catch was simply moved to the outlier longline data

file for which there is low confidence. It was the case for example, of two outliers in the Japanese-LL fleet detected in the south western Indian Ocean during summer of 1974 and 1980. They had extremely high bigeye catch relatively to a small fishing effort, leading to a factor >100 comparatively to the mean CPUE of neighbouring records of the same month. The impact of the correction of fishing effort on detected outliers can be illustrated with the distribution of variance of the CPUE (Fig. 5).

As it can be expected, the correction of fishing effort of outliers proportionally to the mean CPUE of neighbouring values produced a narrower range of variability in CPUE. However, the impact did not spread evenly over time. A maximum number of outliers with very high CPUE values were detected in the Japanese LL fleet during two short periods, in 1954-1958 and 1977-1978 (Fig. 6). The annual mean CPUEs in those periods were higher than in any other years. For the Taiwanese LL fleet there is relatively limited changes in the

CPUE time series with the largest occurring in 1992. The mean annual CPUE in 2012 is the highest of the whole 47 years time series before and after correction (Fig. 6).

#### 3.4.2 Purse seine

There were 3,472 outliers detected in the purse seine dataset. For the largest fleet, i.e., the Spanish-PS-LS-FHOURS (17,586 data) and the French-PS-LS-FHOURS (14,118 data), the outlier threshold value was set to

five to avoid a too selective criteria since variability in purse seine fishing CPUE can be much higher than with longline. With this threshold 5.7% and 5% of the records of respectively the Spanish and French fleets were detected as outliers (Table S1). Fishing efforts of these outliers were corrected with the mean CPUE of neighbouring records in a maximum radius (r) of 5° (the resolution for purse seine data being 1°). For associated log data set (LS), 94.5% of detected outliers were corrected, 31.8% were corrected using mean CPUE within

radius 1°, 36.9% with r = 2°, 20.2% with r = 3°, 6.9% with r = 4°, and 4.1% with r = 5°. Using also a maximum radius of 5 degrees, it was possible to correct 74% of total associated mix fishery's outliers (MIX), 59.6% of total associated free schools' outliers (FS), and 83.3% of uncategorized outliers (UNCL), were corrected. The non-corrected outliers (438 records) were kept in a separate fishery (Purse Seine Outliers).

The impact of this outlier screening and correction on time series CPUE is shown for Spanish and French Log-

associated purse seine fleets on the Fig. 6. Unlike with longline data the effect was more uniformly distributed over time. Despite that the correction concerned only 5% of the data, the change was also stronger than in the case of longline data. For the French fleet however, the difference with the corrected series decreased in the early



2000s and remained after 2005 in its smallest range of deviation. For this fleet, the correction reduced variances of CPUE, particularly in the eastern Indian Ocean (Fig. 7).

### 3.4.3 Others-gears

A total of 564 outliers were detected from the group other gears, from which 346 (61.3%) were corrected using neighbouring points within a maximum radius of 5 degrees (78.6% within a radius of 3°). Non-corrected outliers (219 records) were kept in a separate data file (Other Gear outliers).

### 3.5 Effort unit

Longline fishing efforts were converted to number of hooks from fishing days for the Portuguese-ELL (2006, 2007, 2013, 2014), the Mayotte-ELL, the Philippine-LL, the Thailand-LL (2012, 2013), and from number of sets for the Thailand-LL (2007, 2008). They were converted using mean CPUE ratio for the period available with the reference unit. For the Mayotte-ELL and the Philippine-LL that only provided one year of fishing-days data, the ratio was calculated from respectively the Portuguese-ELL and the Thailand-LL number of hooks fishery.

Three Spanish and two Japanese purse seine fleets required effort standardization. The Spanish efforts were standardized to number of fishing hours. It can be checked that converted efforts of the LS and the FS sub-datasets occupy ranges of the reference efforts (Table S2). The efforts of Japanese and Thailand-LS were standardized to number of fishing days. For other-gears group, efforts of the Maldives coastal longline and the La Reunion hand line and troll line were standardized to number of hooks.

### 3.6 Detected breaks in Japanese longline fishery

From the eight regions (Fig. 2) defined to investigate historical changes in fishing practices of the Japanese longline fishery, the region VII was excluded of the BFAST analysis because of too many missing values during the time periods 1955-1961, 1972-1990, and 2007-2014. The monthly CPUE time series for the seven remaining regions varies between 605 months (~50 years) and 746 months (~62 years). The fishery in the region VIII has the longest series, from November 1952 to December 2014 (Table 2).

For the BFAST parameterization, we tested a minimum duration of time series between 10 and 25 years or a maximum number of breakpoint between one and four. There was no change in the BFAST results for a value above four. The period of ten years was selected because we sought a break corresponding to long-term change of fishing strategy, while the twenty five years corresponds to the maximum length that can be selected to detect at least one break. Detected breaks were considered very robust when they were detected in at least 75% of the tests carried out with the two parameterization approaches.

Based on these thresholds, two very robust time breaks were detected in the north western (region I) and the eastern tropical (region V) Indian Ocean (Table 2 and Fig. 8). They occurred in Oct. 1980 in region I and May 1977 in region V. A third breakpoint was detected in region III in Aug 1977. As a consequence the Japanese longline fishing data of region I, III and V were merged into two historical periods. The first one includes data of the period Mar 1955-Oct1980 for region I with period Nov 1952-May 1977 for region V and Jan 1955-Aug 1977 for region III. The remaining Japanese longline fishing data were aggregated into a single time series of tropical (region II, III, and IV) and sub-tropical (VI, VII, and VII) fisheries (Table 3).





### 3.7 Final definition of fisheries

With the four Japanese longline fisheries (L1 to L4) defined above, two Taiwanese longline fisheries (L5-L6) were defined with a similar spatial stratification between tropical (north of 15° S) and sub-tropical (south of 15° S) regions (Table 3). Other longline fisheries are the Korean-LL (L7) and then the more recent fleets defined on the IOTC criteria: longline (LL), fresh longline (FLL), swordfish longline (ELL), experimental longline (LLEX). A last file (L12) gathers all outlier data for which there is low confidence (Table 3). Eight purse seine fisheries (S13 – S20) were defined based on the fishing strategy and effort unit (Table 3) including also a file for outliers (S20). Finally, ten other fisheries were defined for the other-gears group (fishery O21 – O30). Nine types of gear were assigned in separate fisheries (O21 – O29). The last fishery (O30) is for non-corrected outliers (Table 3).

### 3.7 Length frequency data

The available length-frequency data coincide with the definition of eighteen fisheries. They are: L1-L10, L12, S13, S15, S16, S19, S20, O22, and O29. For the longline fisheries, the highest number of size data is from the tropical Taiwanese-LL (Fishery L5) with 9,583 samples over 1980-2014 and the large fish measured (mean fork length > 135 cm) in the western Indian Ocean (Fig. 9). For the purse seine fisheries, the largest number of samples, 11,433 over 1984-2014, comes from the log-associated fishery (S13). Mean fork-lengths of catch higher than 52 cm in this fishery are distributed over the central and eastern Indian Ocean (Fig. 9).

### 4 Discussion

This study produced a standardized geo-referenced dataset of historical fishing data for the Indian Ocean bigeye tuna based on data collected by the Indian Ocean Tuna Commission (IOTC). While the catch data have been raised to the level of annual nominal catch declared by all IOTC country members, there are still uncertainties and unreported catch for several fleets. This concerns particularly the fresh-tuna longline fisheries from Taiwan-ROC before 2006 and from Indonesia, the industrial purse seine fleet of Iran, the industrial longline fleets of India, Indonesia, Malaysia, Oman and Philippines, and the more artisanal Iranian and Pakistan driftnet fleets or Sri Lanka gill net fleet (IOTC, 2015). Nevertheless, available data were analysed to identify inconsistencies and errors as suggested by Sharma et al. (2014) and Hoyle et al. (2015).

Until now, most stock assessment studies of bigeye tuna conducted by IOTC have been based fishing data aggregated either over the whole oceanic basin or a few large areas (e.g. Kolody et al., 2010; Langley et al., 2013), and geo-referenced data were used mainly for investigating the change in CPUE of a few main fisheries (e.g., Matsumoto et al., 2015; Yeh and Chang, 2015). The comprehensive geo-referenced fishing dataset prepared here allows to envisage future stock assessment studies accounting for more detailed spatial structures, which is a key issue for highly migratory species like tunas (Maunder et al., 2014; Punt et al., 2014; Sharma et al., 2014; Lehodey et al., 2014). In this context, even small domestic fisheries representing a very small portion of catch can provide useful information on the distribution of the species.

The detailed and careful analysis of these fishing data sets has shown several inconsistencies that we tried to resolve on the best possible way. However, it was impossible, to propose a single catch unit for the Asian longline fleets. The catch conversion from number of individuals to total weight produced unrealistic results





when compared to declared annual nominal catch. This conversion relies on the availability of length frequency samples coinciding in time and space with the catch declared in number of fish, or on the co-occurrence of catch declared in weight and number of fish by the same fleet in the same spatio-temporal strata. Unfortunately, length-frequency sampling does not cover the whole fishery activity (Sharma et al., 2014), and at the monthly 5°

square resolution used for longline data, co-occurrences of both catch units represented 45% and 2.8% of the cases for the Japanese and Korean fleets respectively. Thus, it was necessary to decrease the resolution of spatio-temporal strata used for conversion and even to use monthly average or single conversion factors in many cases. The final result being inconsistent with the nominal catch declaration, it was decided to keep unchanged the Japanese and Korean longline catch data using number of individuals.

Nevertheless, before submitting their national annual fishing statistics to IOTC, the Japanese National Research Institute of Far Seas Fisheries (NRIFSF) applies raising procedure to provide geo-referenced data consistent with the nominal catch declaration (Matsumoto et al., 2013). Therefore, the sum of catches of the two Japanese longline geo-referenced datasets, with catch expressed in weight and in number, should be consistent with the total catch declared. The Korean National Fisheries Research and Development Institute (NFRDI) has

aggregated catch from fishermen's logbooks into monthly 5x5 degree cells (Lee et al., 2014), but whether the catch were raised to nominal catch or un-raised is uninformed. Geo-referenced Korean catch declared in weight were raised to nominal catch. For those declared in number of fish, our attempts to convert the catch to weight were unsuccessful, suggesting that the geo-referenced Korean catch expressed in number of fish is underestimated.

As reported by Chassot et al. (2015), raising procedure is also conducted by fishery scientists involved in IOTC statistical working group to match geo-referenced purse seine catch data to the level of nominal catch. This seems to be confirmed by the good match between geo-referenced and nominal catch data that we obtained. This is not the case however, for the European longline fleets (La Reunion, Mayotte and Portugal), for which a raising procedure was applied in this study.

There are various potential sources of mistakes along the chain of fishing data reporting and different approaches to check and screen these data. For instance, Japanese fishery scientists check the effort data of the longline logbooks and remove those with less than 200 or more than 5000 hooks (Hoyle et al., 2015). We employed a robust outlier filtering method based on CPUE to detect anomalous data. Then, instead of removing the catch and effort observation, the fishing effort value was corrected relatively to the nearest neighbour values, to avoid

as far as possible an underestimation of the catch which is the key information for fishing mortality estimates. When it was impossible to correct the fishing effort in absence of neighbouring values, the catch observation was retained in a special fishery (outliers) allowing keeping track of all declared catch.

Among the largest anomaly detected there is a high peak of CPUE in 2003 for the Taiwanese longline fleet that has been already identified in previous analyses and potentially linked to misreporting of logbook data that

occurred among the Taiwanese fleets operating in the Pacific, Atlantic and Indian Oceans (see Hoyle et al., 2015). Unusual very high CPUE observed in the Spanish and French log-associated purse seine sets were detected in 1999. Since this was observed in both fleets, it is likely that this particular year was effectively highly favourable. Despite a threshold value set to five in the Hampel Identifier method for these purse seiner data, a substantial number of effort data were classified as outliers and corrected. It is possible that these high peaks in

CPUE variability reflect some heterogeneity in the fleets, e.g. due to the few super-seiners (> 2 000 gross





tonnages) and super super-seiners (> 3 500 gross tonnages) used by Spanish and French fleets (Lopez et al., 2014; Davies et al., 2014).

While the number of hooks seems a reliable measure of fishing effort for passive fishing gears like longlines, it is much more difficult to define consistent fishing effort unit for purse seiners. The time spent for searching tuna
schools can be highly variable depending on the skills of the skipper, the technology used, the engine power, and the communications between boats. By using only fishing hours, the effort unit is supposed to be independent of such variability, though there is still some uncertainty on what is included in this time of fishing activity. The effort of French purse seiners of the geo-referenced IOTC data set was already standardized entirely to number of fishing hours through re-processing of data for the period 1981-1990 when effort were not declared with this
unit (Chassot et al., 2013). For the Spanish fleets we similarly converted the effort to number of fishing hours for the period 1984-1990 to have homogeneous series based on the same unit. The comparison of both series in fishing hours showed that the French fleet had a lower annual total effort than the Spanish fleet except for the beginning of the fishery between 1981 and 1988. This is consistent with the number of purse seine vessels of both fleets operating during these years, ranging between 21-26 and 12-21 for French and Spanish fleets
respectively until the mid-1980s but increasing to 26 for the Spanish fleet during 1989-1990 (Pianet et al., 2008). Over long historical periods, a fishery is potentially subject to strong changes due to exploitation, market and technological evolutions. The Japanese longline fleet has been the most important bigeye tuna fishery in the Indian Ocean. It provides the longest time series since the early 1950s that has major influence on all stock assessment studies. Important changes have been documented for this fleet. Until the mid-1950s, the fleet was
still limited to the Eastern Indian Ocean (south off Java). Then the fishing ground expanded into the central and western tropical Indian Ocean (Mohri and Nishida, 1999). In the 1970s and 1980s, with an increasing demand for the sashimi market, the introduction of monofilament allowed to set the line in deeper depth to target bigeye tuna. This produced major changes in the catchability of this species (Campbell et al., 2001; Okamoto et al., 2001; Ward and Hindmarsh, 2007; Hoyle et al. 2015). The detailed analysis of CPUE by geographical strata
conducted in the present study allowed identifying the timing of change. Consistent and robust breakpoints were identified in the western and eastern tropical Indian Ocean. The break is detected earlier in the eastern than in the western tropical regions. This is confirmed by Okamoto et al. (2001) who reported that the use of deep tuna longline started in the southern of Java and western of Sumatra around 1977 and then extended to the western equatorial Indian Ocean.
Finally, a total of thirty fisheries were defined to cover the whole period of exploitation of bigeye tuna in the Indian Ocean since 1952, with their associated catch length frequency data. There are certainly further sub-disaggregation possible to get still more homogeneous fisheries data sets, but it is necessary to find a balance with a reasonable number of fisheries that can be manipulated in further studies of complex spatial fish dynamics. It could be also possible to merge a few of these fisheries in the future, particularly if the conversion
from individual to weight can be reconstructed from original Japanese and Taiwanese longliners logbooks that have finer spatial resolution than IOTC's datasets (Hoyle et al., 2015). If necessary the number of fisheries could be also limited to the main fleets that over the historical period of exploitation extracted most of the bigeye tuna catch. For instance, the ten first longline fisheries (L1-L10) together with the four first purse seine fisheries (S13-S16) represent 96 % of the total bigeye tuna catch in the Indian Ocean during 1952-2014 (Table 3).



In summary, the dataset proposed here is thought to be a practical and useful geo-referenced representation of the historical distribution of bigeye catch over its modern history of exploitation. Hopefully, new communication technologies, e.g., the Electronic Report System (ERS), should help in the improvement of fishing data statistics. Nevertheless, strong networks of observers and port samplers will continue to be additional requisite to monitor

these fisheries and provide the most critical information for assessing their impact on the stocks.

**Acknowledgements**

We are grateful to IOTC for the access to its public fishing database. This work was supported by the Infrastructure Development of Space Oceanography (INDESO) project. The first author would like to thank Roger de Jong for providing codes of BFAST's validation.

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





**Tables**

**Table 1: Differences between nominal and geo-referenced total catch of bigeye tuna in the Indian Ocean for fishing fleets that required a data raising procedure.**

| Fleet | Time Period | Total nominal catch (tonnes) | Difference with geo-referenced catch (%) |
|---|---|---|---|
| Korea-LL (tonnes) | 1994-1997, 1999-2008,2010, 2011 | 57,001 | 50.1 |
| Mauritius-LL | 2001, 2003-2010 | 120 | 19.7 |
| Seychelles-LL | 2000, 2001, 2003-2014 | 66,121 | 17.6 |
| La Reunion-LL | 1994-2014 | 5,414 | 49.4 |
| South Africa-ELL | 1998-2014 | 2,056 | 34.5 |
| Mayotte-ELL | 2005 | 23 | 27.6 |
| Portuguese-ELL | 2006-2008, 2010, 2011, 2013, 2014 | 484 | 25.6 |
| Australia-ELL | 1992-2014 | 3,807 | 24.5 |
| Mauritius-ELL | 2001-2006, 2010-2013 | 76 | 22.1 |
| Seychelles-ELL | 1983-1985, 1995-1999, 2001-2014 | 1,043 | 11.5 |
| Taiwanese-FLL | 2010-2014 | 18,270 | 67.0 |
| Malaysia-FLL | 2013,2014 | 92 | 23.5 |
| Mauritius-PS-LS | 1989-1995, 1997-1999 | 1,314 | 37.8 |
| NEISU-PS-LS | 1992-1994, 1998-2002 | 10,140 | 6.1 |
| Mauritius-PS-MIX | 1988-2000 | 6,304 | 40.5 |
| Mauritius-PS-FS | 1988-1991, 1993, 1994, 1997-1999 | 204 | 54.8 |
| Indonesia-PSS-UNCL | 1986 | 1,269 | 99.4 |
| Taiwan-GILL | 1986-1991 | 2,851 | 6.9 |
| Sri Lanka-GL | 1994-2006 | 719 | 49.6 |
| South Africa-SPOR | 1991, 1992,2012 | 52 | 96.6 |
| Indonesia-TROL | 1988 | 249 | 99.7 |



**Table 2: Results of the BFAST analysis of Japanese time series. Breakpoints with at least 50% detections in both parameterization approaches are highlighted with bold letters.**

| Region | Period | Detected time breaks from period of time series 10-25 years (% of detection) | Detected time breaks from breakpoint fixed to 1-4 (% of detection) |
|---|---|---|---|
| I | Mar-55 to Nov-07 | **Oct-80 (100)** | **Oct-80 (100)** |
| II | Mar-54 to Dec-09 | Feb-74 (6); Sep-77 (13); Oct-77 (63); May-82 (13); Feb-87 (6); Jul-97 (13) | Feb-74 (13); Sep 77 (6); Oct 77 (6); Jan-87 (13); Jul-97 (19) |
| III | Jan-55 to Dec-10 | Nov-72 (6); Sep-73 (6); Oct-76 (6); Apr-77 (6); **Aug-77 (50)**; Sep-80 (19); Sep-90 (6); Apr-92 (38); Oct-92 (1) | **Aug-77 (100)**; Oct-86 (13); Jun-95 (13) |
| IV | Mar-54 to Dec-10 | Oct-70 (38); Oct-71 (6); Aug-81 (50); Mar-88 (38); Mar-90 (6) | Oct-70 (75); Aug 81 (6); Mar-88 (19) |
| V | Nov-52 to Dec-10 | **May-77 (88)**; Nov-77 (6) | **May-77 (100)** |
| VI | Aug-60 to Dec-10 | Dec-89 (6); Dec-90 (6); May-91 (13); **Jun-93 (69)** | Aug-68 (50); Feb-69 (25); Sep-76 (50); May-91 (50); **Jun-93 (50)** |
| VIII | Nov-52 to Dec-14 | Mar-67 (25); Feb-68 (6); Jun-69 (6); Oct-69 (6); Dec-83 (50); Dec-84 (25); Oct-92 (13); Feb-93 (6); Mar-00 (6); Mar-01 (6); Mar-02 (13) | Mar-67 (13); Feb-67 (13); Nov-76 (13); Nov-86 (13); Feb-93 (6); Feb-05 (6) |





Table 3: Final definition of historical bigeye tuna fisheries in the Indian Ocean. Codes: LL=longline targeting bigeye, FLL=fresh tuna longline, ELL=longline targeting swordfish, LLEX=exploratory fishing longline, PS-LS=purse seine sets on associated logs, PS-FS=purse seine sets on free schools, PS-MIX=purse seine with mixed strategy, PSS=small purse seine, GILL=gillnet, GL=combination gillnet and longline, HATR=hand line and troll line, LLCO=coastal longline, UNCL=unknown gear, BB=baitboat, HAND=hand line, SPOR=sport fishing, TROL=troll line.

| Code | Flag | Gear | Catch Unit/Effort Unit | Period | Nb Data | Nominal Catch (%) | Res. |
|---|---|---|---|---|---|---|---|
| L1 | Japan region I, III &V | LL | nb.individual/nb. hooks | 1952-1980 | 5,526 | 4.76 | 5 |
| L2 | Japan region I, III &V | LL | nb.individual/nb. hooks | 1977-2014 | 6,459 | 6.13 | 5 |
| L3 | Japan region II & IV | LL | nb.individual/nb. hooks | 1954-2014 | 8,048 | 6.31 | 5 |
| L4 | Japan region VI, VII & VIII | LL | nb.individual/nb. hooks | 1952-2014 | 11,868 | 6.78 | 5 |
| L5 | Tropical Taiwan | LL | tonnes/nb. hooks | 1967-2014 | 15,721 | 31.55 | 5 |
| L6 | Sub-tropical Taiwan | LL | tonnes/nb. hooks | 1967-2014 | 9,190 | 3.29 | 5 |
| L7 | Korea | LL | nb.individual/nb. hooks | 1975-1987, 1992, 1993, 2009, 2012, 2013, 2014 | 6,838 | 9.25 | 5 |
| L8 | Korea, China, Seychelles, Mauritius, Thailand, Maldives, Philippines | LL | tonnes/nb. hooks | 1994-1997, 1999-2014 | 7,453 | 6.17 | 5 |
| L9 | China, Taiwan, Malaysia | FLL | tonnes/nb. hooks | 2006-2014 | 1,958 | 0.65 | 5 |
| L10 | Seychelles, Australia, La Reunion, South Africa, Mauritius, Mayotten, Portugal | ELL | tonnes/nb. hooks | 1983-1985, 1992-2014 | 3,321 | 0.41 | 5 |
| L11 | India | LLEX | tonnes/nb. hooks | 1991, 1995-1997, 2007, 2009, 2011-2013 | 40 | 0.02 | 5 |
| L12 | Outliers of longline | LL, ELL ,FLL ,LLEX | tonnes/nb. hooks, tonnes/nb. fishing days | 1953-1955, 1957,1958, 1961,1963-1968,1970, 1971,1973-1986,1989, 1991-2014 | 214 | 2.3 | 5 |
| S13 | France, NEIPS, Spain, Mauritius, Seychelles | PS-LS | tonnes/nb. fishing hours | 1981-2014 | 45,738 | 11.25 | 1 |
| S14 | France, Spain, NEIPS, Mauritius, Seychelles | PS-MIX | tonnes/nb. fishing hours | 1981-2014 | 9,745 | 6.61 | 1 |
| S15 | France, NEIPS, Spain, Mauritius, Seychelles | PS-FS | tonnes/nb. fishing hours | 1981-2014 | 7,452 | 1.72 | 1 |
| S16 | Soviet, Japan, NEISU, Thailand | PS-LS | tonnes/nb. fishing days | 1986-2014 | 3,866 | 1.04 | 1 |
| S17 | Japan | PS-MIX | tonnes/nb. days at sea | 1986,1989-1997, 1999,2007 | 2,287 | 0.53 | 1 |
| S18 | Japan, NEISU, Thailand, Korea | PS-FS, PS-MIX, PS-LS | tonnes/nb. days at sea, tonnes/nb. fishing days, tonnes/nb. sets | 1989-1994, 1997-2002, 2006, 2009, 2012-2014 | 997 | 0.25 | 1 |
| S19 | Indonesia | PSS | tonnes/nb. trips | 1986 | 63 | 0.04 | 1 |
| S20 | Outliers of purse seine | PS | tonnes/nb. fishing hours, tonnes/nb. fishing days, | 1984-2014 | 438 | 0.63 | 1 |





| Code | Flag | Gear | Catch Unit/Effort Unit | Period | Nb Data | Nominal Catch (%) | Res. |
|------|------|------|------------------------|--------|---------|-------------------|------|
| | | | tonnes/nb. days at sea, tonnes/nb. Dets | | | | |
| O21 | Taiwan | GILL | tonnes/nb. net-days | 1986-1991 | 6,508 | 0.1 | 1 |
| O22 | Sri Lanka | GL | tonnes/nb. trips | 1994-2006 | 2,100 | 0.03 | 1 |
| O23 | La Reunion | HATR | tonnes/nb. hooks | 2005, 2006, 2011, 2012 | 1,025 | 0.01 | 1 |
| O24 | Maldives | LLCO | tonnes/nb. hooks | 2013, 2014 | 597 | 0.09 | 1 |
| O25 | Sri Lanka | UNCL | tonnes/nb. trips | 1994-1998, 2002-2004 | 444 | 0.03 | 1 |
| O26 | Australia, Maldives | BB | tonnes/nb. days at sea, tonnes/nb. fishing days | 1994,1997, 2013, 2014 | 120 | 0.03 | 1 |
| O27 | Maldives | HAND | tonnes/nb. fishing days | 2014 | 83 | 0.02 | 1 |
| O28 | South Africa | SPOR | tonnes/nb. days at sea | 1991, 1992 | 50 | 0.01 | 1 |
| O29 | Indonesia | TROL | tonnes/nb. trips | 1988 | 30 | 0.01 | 1 |
| O30 | Outliers of other-gears | GILL, GL, UNCL, HATR, BB | tonnes/nb. net-days, tonnes/nb. trips, tonnes/nb. boats, tonnes/nb. hooks, tonnes/nb. fishing days | 1988-1990, 1994-1996, 2005, 2011, 2014 | 219 | 0.06 | 1 |



**Figures**

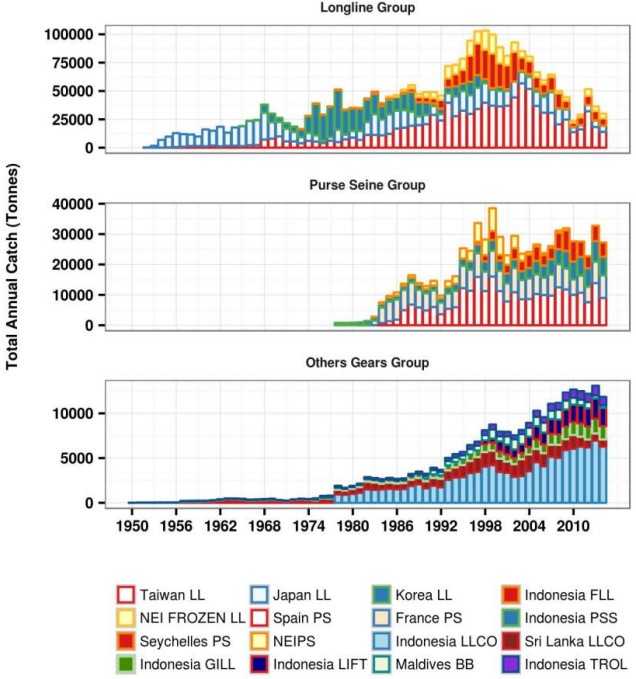

Figure 1: Total annual bigeye nominal catch in the Indian Ocean caught by longline targeting bigeye tuna (LL), fresh tuna longline (FLL), purse seine (PS), small purse seine (PSS), coastal longline (LLCO), gillnet (GILL), liftnets (LIFT), baitboat (BB), and troll lines (TROL). To enhance visibility, only fleet contributing to catch larger than 5% of total catch in each fishing gear group are shown.

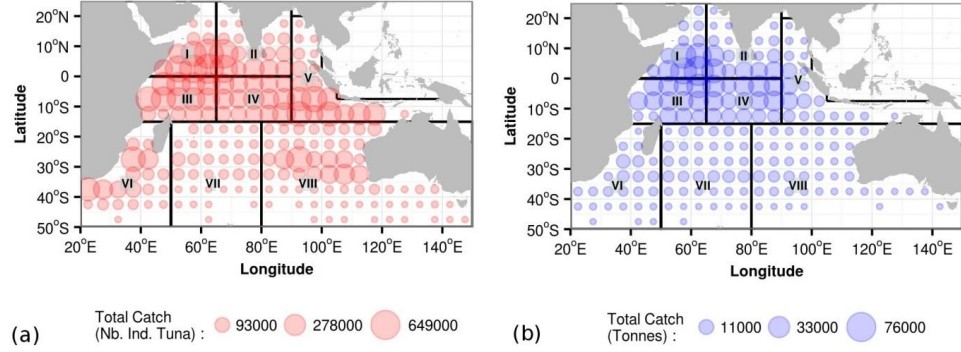

Figure 2: Spatial distribution of bigeye tuna catch by longline fishing gears (Total catch over 1952-2014. (a) Catch from the Japanese and Korean fleets expressed in number of individual tuna; (b) Catch from the remaining longline fleets expressed in metric tonnes.



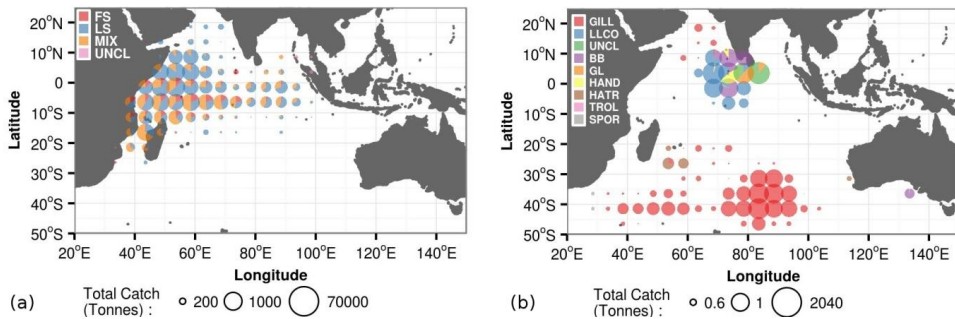

**Figure 3: Spatial distribution of bigeye tuna catch by (a) purse seine fishing gears (1981-2014) and (b) other gear group (1986-2014). Codes: FS=free schools sets, LS=associated logs, MIX=mixed strategy, UNCL=unknown purse seine sets, GILL=gillnet, LLCO=coastal longline, UNCL=unknown gear, BB=baitboat, GL=gillnet longline combination, HAND=hand line, HATR=hand line and troll line, SPOR=sport fishing.**

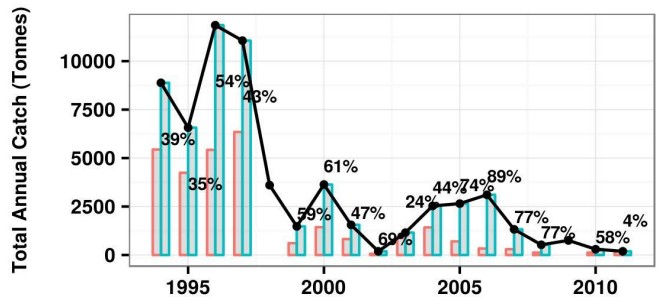

**Figure 4: Total annual nominal catch (black line) and total annual geo-referenced catch (bars) of Korean longline (tonnes sub-dataset). Numbers associated to the bars indicate percentages of annual missing catch before raising.**



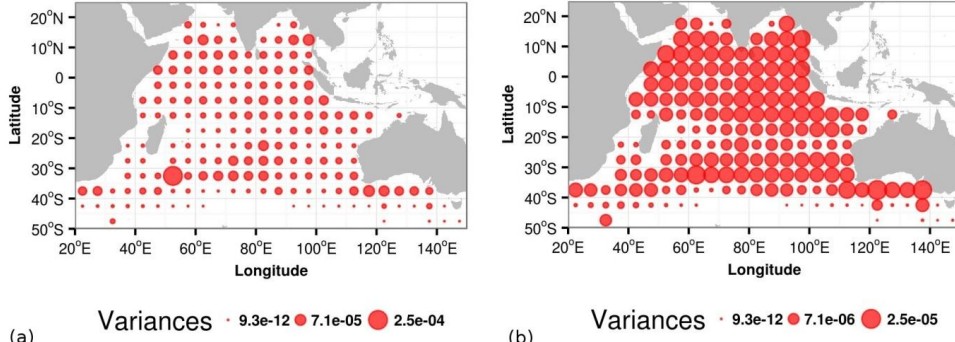

**Figure 5: Spatial variances of catch per unit effort (CPUE) computed from the Japanese longline fleet before (a) and after correcting or eliminating outliers (b).**

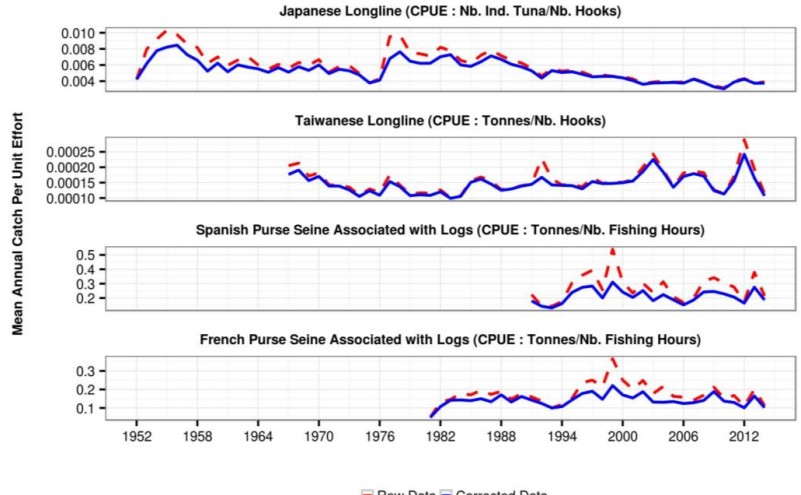

**Figure 6: CPUE time series before (dashed red line) and after (solid blue line) correction of efforts of outliers for the Japanese, Taiwanese longline, the Spanish and French purse seine fleets fishing on logs (LS).**



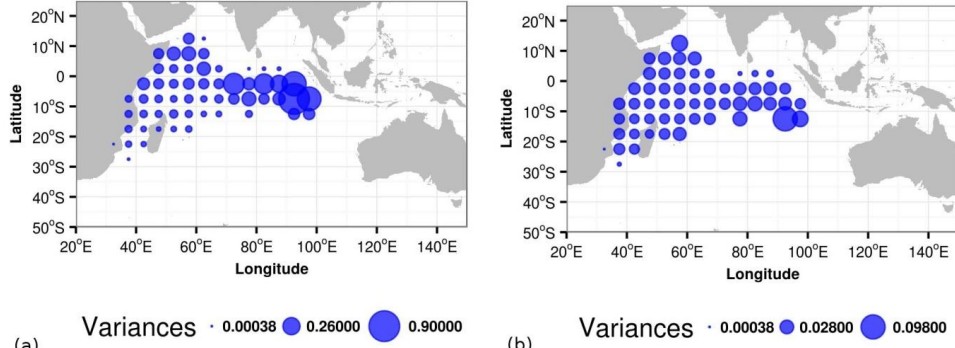

**Figure 7: Spatial variances of catch per unit effort (CPUE) computed from French log associated purse seine before (a) and after correcting or eliminating outliers (b). To enhance visibility, the variances were calculated for each 5x5 degree cell.**

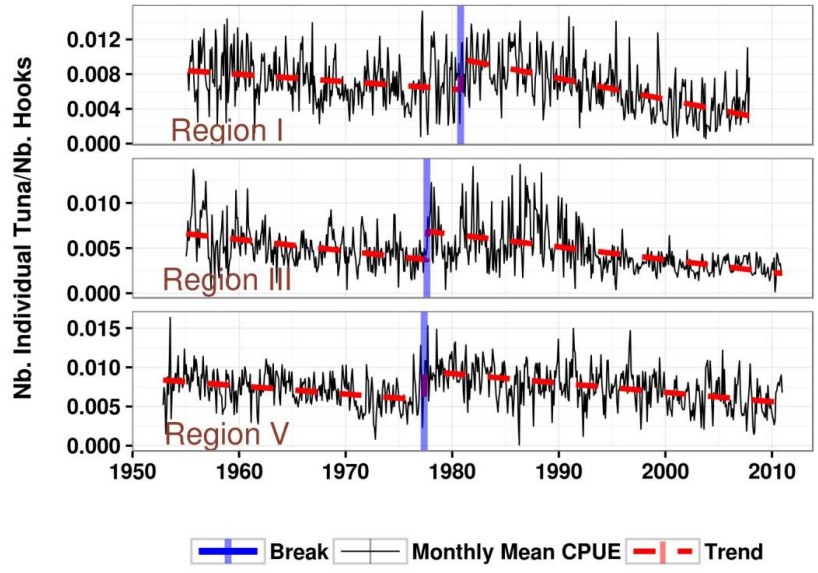

**Figure 8: Time series monthly CPUE, trend and detected break over the western (Region I and II) and the eastern tropical Indian Ocean (Region V).**



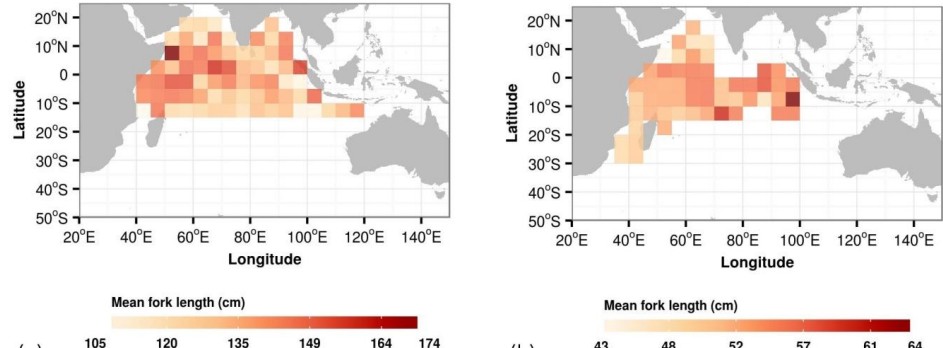

**Figure 9: Distributions of bigeye tuna size derived from catch sampling in (a) the tropical Taiwanese longline fishery (L5), and (b) the purse seine log-associated fishery (S13). The original data of size frequency from both fisheries are spatially distributed in 5x5 degree cells.**