# Peer review of "Standardization of a geo-referenced fishing dataset for the Indian Ocean Bigeye Tuna, *Thunnus obesus* (1952-2014)"

_Earth System Science Data, 2016_

## Referee Comment (RC1) · A.W. Diamond (Referee) · 20 Oct 2016

My expertise in this area is very limited; I suspect I was asked to review because of my work on seabirds in the Indian Ocean in the past, including identifying tuna as critically important in making prey available to seabirds fishing at the surface of tropical oceans. I therefore have an interest in seeing trends in the tuna fishery as they might affect the ability of many species of seabirds, especially frigatebirds, terns and boobies, to obtain sufficient food. The work presented here is clearly very important in providing consistent data from many datasets of variable quality, and I commend the authors for attempting to do this. I do suggest a few concepts could be better explained for the non-expert reader - if there are any such in this journal! I did not clearly understand the

concept of "raising catch data" to "the nominal catch"- could the authors explain just what is a nominal catch, and why it is necessary to raise the existing data to it? The disparity between the two (Table 1 and Figure 4) is so large that it reduces confidence in the reliability of the data resulting after the complex adjustments that the authors have made. Nonetheless, the steep increase in tuna catches over the time covered (Figure 1) is dramatic and important; thus it is particularly reassuring to see from Figure 6 that CPUE has not declined as much as might be expected, except in the Japanese longline fleet.

I have a few suggestions for improving the English which are given below.

p.1 L 27 delete 'of' between Eastern and Indian

L 32 'of' should be 'for'

p.2 l 15 delete 'to'

p.3 l 33 Need to say what was the cutoff length (or weight) between 'large' and 'small' purse-seiners.

p.5 l 24 change 'to be provided' to 'being provided'

p.9 l 15, delete "it"

p.11

l.23 This is the first mention of Iran, India, Pakistan and Oman! l. 26 insert "on" between "based" and "fishing"

p.13 l. 4 "for" should come after "searching"

---

## Referee Comment (RC2) · Anonymous Referee #2 · 25 Oct 2016

I read this paper with interest. Authors have gone to great length to explain the process of modifying the data and raising it to nominal catch. The nominal catch is not well explained and I believe that this needs greater clarification. Many of the data sets have been subject to many filtering and analytical algorithms, but the text is difficult to follow because many aspects are not clear. For example, the authors refer to a 'robust outlier filtering method was used without citing any particular method. Thus it may raise questions in how the data was modified in some cases. There needs to be themes within the introduction and discussion to allow the reader to visualize where the authors are going with this. The Discussion also does not do justice. It only identifies a few weaknesses in the data. I was hoping to see some comparative

analysis highlighting where the discrepancies arise between the reported catch and the standardized catch is not well explained (although it is evident in the figures and tables). I advise the authors to highlight this more in the discussion and propose a path forward. I feel that the paper is an important contribution in data standardization, but part of the paper need to be further clarified to allow readers to better appreciate its value.

---

## Editor Comment (EC1) · F. Huettmann (Editor) · 9 Nov 2016

Dear Authors, thanks for this insightful data set and its description and overview.

As an editor, if I may, I have one tiny comment to add please, and that is, ISO compliant metadata.

I lack seeing and reading about them in this contribution.

Checking the PANGAEA database and DOI, I see no reference made to Metadata, nor does the MS include them.

There is a standard way of providing metadata, e.g. ISO compliant ones with 400

entries, and I kindly ask the authors to address it. Namely, details for each column in the database and the spatial geographic datum and coordinate units. If the data is served in ArcGIS formats (as the maps imply to me) that would be a good format to explain as well (ideally, ASCII or move into OpenGIS like QGIS or SAGA or R) and its metadata.

Again, ideally, this detail should come right with the data themselves, as per ISO meta-data in XML. If that is not possible, then please provide in the Appendix, or Methods even.

Thanks for the consideration.

This data set will add a great deal to our understanding, use, research and management of thuna in the Indian Ocean and is really worthwhile the efforts.

Thanks again; very best Falk Huettmann PhD, Associate Editor

---

## Author Comment (AC1) · 24 Nov 2016

1. Concept of raising catch data to the nominal catch data:

Response: The introduction was revised to give a more detailed explanation: "The nominal catch data set is the official annual catch declaration by each Country Member to the Commission. It gives total annual catch and fishing effort by species and by fishing gear, usually sub-divided by month. However, there is no geo-referenced information on where the fish are caught. This information is partially provided in the second dataset that gives sub-samples of monthly geo-referenced catch and effort by fleet".

[Figure]

2. The disparity between the two (Table 1 and Figure 4) is so large that it reduces confidence in the reliability of the data resulting after the complex adjustments that the authors have made.

Response: Since the geo-referenced catch are subsets of the nominal catch, it is necessary to use a raising procedure. There is no other choice to obtain a spatially explicit distribution of total catch allowing to account for all fishing mortality in studies requiring geo-referenced fishing data. We use the most accurate approach for conversion from number of fish to weight and for raising to nominal catch, by using available size frequency samples of catch with the best possible match between fishery, location and date. For the main fisheries that have generally large enough sub-sampled geo-referenced data of both catch and length frequencies of catch, the result is very consistent. A good example is provided with a new figure (Fig. 1) comparing nominal catch with total catch from geo-referenced dataset after conversion from individual number of fish to weight using available samples of size frequencies of catch. For the main Japanese longline fishery, the difference is only ∼11 %. The largest uncertainty is essentially for small fisheries but thus concerning a small amount of catch.

3. p.1 L 27 delete 'of' between Eastern and Indian. Response: done

4. L 32 'of' should be 'for'. Response: done

5. p.2 l 15 delete 'to'. Response: done

6. p.3 l 33 Need to say what was the cutoff length (or weight) between 'large' and 'small' purse seiners.

Response: The first sentence of the subsection "2.1.2 Purse seine" was modified as follow: "Geo-referenced purse seine fishing data was divided between large (PS) and small purse seine (PSS). The PS has carrying capacity about 1,000 to 1,500 tonnes, while the PSS has less then about 200-250 tonnes (Joseph, 2003). The PS consists of geo-referenced data from fleets of Spain, France, Seychelles, Japan, Mauritius, Thailand, Korea, former Soviet Union, NEISP and NEISU.

7. p.5 l 24 change 'to be provided' to 'being provided'. Response: done

8. p.9 l 15, delete "it". Response: done

9. p.11 l.23 This is the first mention of Iran, India, Pakistan and Oman.

Response: The original sentence is rephrased and moved to perspective subsection. The new sentence is follow: ". There are some uncertainties that are described and need to be accounted for when using these data. The uncertainty of fishing mortality for certain fleets due to unreported geo-referenced catch should be addressed in future datasets. Catch monitoring in some Countries has been long to implement or is still inexistent, especially for artisanal fleets that however may contribute to a substantial catch due to a large number of small boats. This is likely the case for the artisanal Iranian and Pakistan driftnet fleets or the Sri Lanka gill net fleet (IOTC, 2015), the purse seine fleet of Iran and from distant-water longline fleets of India, Indonesia, Malaysia, and Philippines.."

10. p11 l. 26 insert "on" between "based" and "fishing". Response: done

11. p.13 l.4 "for" should come after "searching". Response: done
* * *
**Japanese Longline: Missing catch = 11%**

**Korean Longline: Missing catch = 40.7%**

**La Reunion Swordfish Longline: Missing catch = 45.1%**

**Total Catch (Tonnes)**

1952 1956 1960 1964 1968 1972 1976 1980 1984 1988 1992 1996 2000 2004 2008 2012

— Nominal Catch    ▮ Geo-referenced Catch

**Fig. 1.** Total annual converted weight catch (red bars) and total annual nominal catch (solid blue line).

[Figure]

---

## Author Comment (AC2) · 24 Nov 2016

1. The nominal catch is not well explained and I believe that this needs greater clarification.

Response: See the response for the 1st reviewer concerning the same point. In addition, the detailed explanation of nominal versus geo-referenced catch is followed by a clarification of the main objective of the study: "A key objective of the present study, by crossing all available information, is to build a geo-referenced dataset, i.e., with monthly catch spatially distributed, that matches the total (nominal) catch for fleets of fishing Countries providing both type of catch, taking into account as far as possible the size selectivity of the fishing gears. "

[Figure]

2. The authors refer to a 'robust outlier filtering method was used without citing any particular method.

Response: The outlier filtering method is described in the method section "2.5 Detection and correction of outliers" . It is the Hampel Identifier method and a reference is provided (Pearson, 1952). The name of the method and its reference is reminded in the discussion.

3. Many of the data sets have been subject to many filtering and analytical algorithms, but the text is difficult to follow because many aspects are not clear. [. . .] There needs to be themes within the introduction and discussion to allow the reader to visualize where the authors are going with this. The Discussion also does not do justice. It only identifies a few weaknesses in the data. I was hoping to see some comparative analysis highlighting where the discrepancies arise between the reported catch and standardized catch is not well explained.

Response: The introduction and discussion have been fully revised to clarify the objectives and methods. Themes have been added in the discussion that is now structured after an introductory paragraph into 3 sections: 4.1 Catch / 4.2 Data screening / 4.3. Fishing effort and catchability / 4.3 Perspectives

Fiinally, we provide a revised dataset of geo-referenced catch, effort and length frequencies for the PANGEA database that includes both the original catch data in number of individuals and their conversion in weight raised to the nominal catch level for the Japanese and Korean longline fisheries as described in Table 3.